# Cdc48/p97 promotes degradation of aberrant nascent polypeptides bound to the ribosome

**Rati Verma[1,2]\*, Robert S Oania[1,2], Natalie J Kolawa[1], Raymond J Deshaies[1,2]\***

[1]Division of Biology, California Institute of Technology, Pasadena, United States; [2]Howard Hughes Medical Institute, California Institute of Technology, Pasadena, United States

**Abstract** Ubiquitin-dependent proteolysis can initiate at ribosomes for myriad reasons including misfolding of a nascent chain or stalling of the ribosome during translation of mRNA. Clearance of a stalled complex is required to recycle the ribosome for future use. Here we show that the ubiquitin (Ub) pathway segregase Cdc48/p97 and its adaptors Ufd1-Npl4 participate in ribosome-associated degradation (RAD) by mediating the clearance of ubiquitinated, tRNA-linked nascent peptides from ribosomes. Through characterization of both endogenously-generated and heterologous model substrates for the RAD pathway, we conclude that budding yeast Cdc48 functions downstream of the Ub ligases Ltn1 and Ubr1 to release nascent proteins from the ribosome so that they can be degraded by the proteasome. Defective RAD could contribute to the pathophysiology of human diseases caused by mutations in p97.

**\*For correspondence:**
verma@caltech.edu (RV);
deshaies@caltech.edu (RJD)

**Reviewing editor**: Ivan Dikic, Goethe University, Germany

## Introduction

Maintaining protein homeostasis is key for the healthy lifespan of the organism (*Koga et al., 2011*) and cells have evolved sophisticated mechanisms to control the quality of newly synthesized proteins (*Bukau et al., 2006*; *Smith et al., 2011*). Dysfunction of these quality control (QC) pathways can tip the balance towards the accumulation of toxic proteins and manifestation of disease, most notably neurodegeneration (*Powers et al., 2009*; *Hartl et al., 2011*; *Dennissen et al., 2012*).

There are multiple pools of QC substrates in cells. A relatively well-characterized substrate pool arises from misfolding of nascent chains that have been properly translated. Depending upon the relative rate at which malfolded sequences that comprise degrons form and bind their Ub ligase receptors vs the rate at which translation is completed, ubiquitination and degradation of misfolding proteins may initiate on the ribosome. It has been shown, for instance, that the acetylated N-terminus recognized by the Ub ligase Doa10 (*Hwang et al., 2010*) is generated co-translationally (*Gautschi et al., 2003*; *Polevoda et al., 2008*), and that the Ub ligase Ubr1 can initiate degradation of an N-end rule substrate prior to completion of its translation (*Turner and Varshavsky, 2000*).

A second pool of QC substrates arises from failure to complete translation successfully. This can occur because the mRNA is defective or the ribosome stalls due to either irresolvable structures in the mRNA or interactions between the nascent peptide and the ribosome (*Lu and Deutsch, 2008*; *Kramer et al., 2009*; *Ingolia et al., 2011*). Defective mRNAs are recognized by a cytosolic mRNA surveillance machinery that mediates their destruction. There are at least three pathways: no-go decay (NGD) targets mRNAs with strong stalls in translation elongation leading to endonucleolytic cleavage of the mRNA, nonsense-mediated decay (NMD) targets messages containing premature termination codons (PTC), and non-stop decay (NSD) targets mRNAs that lack a stop codon. Within this broad scheme, there are many variations by which aberrant mRNAs are generated (*Parker, 2012*). Interestingly, recent data

**eLife digest** Ribosomes are complex molecular machines that translate the sequence of bases in a messenger RNA (mRNA) transcript into a polypeptide that subsequently folds to form a protein. Each ribosome is composed of two major subunits: the small subunit reads the mRNA transcript, and the large subunit joins amino acids together to form the polypeptide. This process stops when the ribosome encounters a stop codon and releases the completed polypeptide.

It is critical that cells perform some form of quality control on the polypeptides as they are translated to prevent a build up of incomplete, incorrect or toxic proteins in cells. Problems can occur if a ribosome stalls while translating the mRNA transcript, or if the mRNA transcript is defective. For example, most mRNA transcripts contain a stop codon, but some do not, and these non-stop mRNA transcripts result in a non-stop polypeptide that remains tethered to the ribosome. It is important that the cell identifies and removes these faulty polypeptides so as to leave the ribosome free to translate other (non-faulty) mRNA transcripts. A regulatory protein called ubiquitin is responsible for marking and sending proteins that are faulty, or are no longer needed by the cell, to a molecular machine called the proteasome, where they are degraded by a process called proteolysis. In 2010 researchers identified Ltn1 as the enzyme that attaches ubiquitin to non-stop proteins in yeast.

Now, building on this work, Verma et al. identify additional proteins involved in this process. In particular, an ATPase enzyme called Cdc48 (known as p97 or VCP in human cells) and two co-factors—Ufd1 and Npl4—promote release of the ubiquitinated non-stop polypeptides from the ribosomes, thus committing the marked polypeptide to destruction by the proteasome. Verma et al. also show that the Cdc48-Ufd1-Npl4 complex is involved in other aspects of quality control of newly synthesized proteins within cells. Collectively these processes are known as ribosome-associated degradation.

Mutations of the gene that codes for human p97 can cause a number of diseases, including Paget's disease of the bone and frontotemporal dementia, so an improved understanding of ribosome-associated degradation could provide new insights into these diseases.

suggest an intimate coupling between QC of defective mRNAs and the nascent peptides they produce (*Shoemaker and Green, 2012*). For example, Upf1 is a positive effector of NMD that is also needed for proteasome-mediated degradation of the prematurely terminated products (*Takahashi et al., 2008*; *Kuroha et al., 2009*).

Quality-control of mRNAs that lack a stop codon and the proteins produced from them has been the focus of considerable recent attention. Translation of these 'non-stop' mRNAs yields a ribosome-tethered polypeptide that is targeted for ubiquitination and degradation (*Ito-Harashima et al., 2007*; *Dimitrova et al., 2009*; *Bengtson and Joazeiro, 2010*). Notably, the mRNA may be processed by multiple pathways (e.g., NSD and NGD) because absence of a stop codon can cause translation to stall either because the end of the message is encountered or polylysine encoded by the poly(A) tail inter-acts with the negatively charged exit tunnel of the ribosome. Non-stop mRNAs can arise naturally through premature termination or polyadenylation within ORFs, which are estimated to occur for up to 10% of all transcriptional events (*Frischmeyer et al., 2002*). Proteins encoded by these messages could potentially impose a considerable burden on QC pathways. Thus, disposal of proteins produced from non-stop messages is likely to be of general significance to understanding human diseases that result from defects in protein homeostasis.

Stalled ribosomes that trigger NGD/NSD are recognized by the Dom34–Hbs1 complex which is structurally related to the termination factors eRF1 and eRF3 but lacks the residues necessary for both stop codon recognition and hydrolysis of peptidyl-tRNA and instead binds to the empty A site in the stalled ribosome (*Doma and Parker, 2006*; *Becker et al., 2011*; *Tsuboi et al., 2012*). Dom34–Hbs1 works in concert with a member of the ABC family of ATPases, known as ABCE1 in mammalian cells (*Pisareva et al., 2011*) or Rli1 in budding yeast (*Shoemaker et al., 2010*; *Shoemaker and Green, 2011*), to disassemble stalled ribosomes for subsequent reuse. Additionally, the translational GTPase Ski7, which is closely related to Hbs1 (and thus to eRF3), has been proposed to recognize ribosomes stalled at the 3′ end of non-stop messages (*van Hoof et al., 2002*). However, no Dom34/eRF1-like

molecule has been identified to date that can interact with Ski7 specifically so it remains unknown if it can promote recycling of ribosome subunits. Following dissociation of subunits, the mechanism by which the associated nascent peptide is extricated remains poorly understood. What has been shown so far is that the Ub ligase Ltn1 can bind the ribosome and ubiquitinate NSD substrates that contain a polylysine tract (*Bengtson and Joazeiro, 2010*). Ltn1 belongs to the RING domain family of E3 Ub ligases that ubiquitinate substrates targeted for degradation by the 26S proteasome (*Deshaies and Joazeiro, 2009*; *Finley, 2009*). However, the mechanism by which the nascent non-stop peptide is released from the ribosome and degraded remains unknown. Nevertheless, the underlying biochemistry is likely to be important for human health, because point mutations in mouse Ltn1 result in neuro-degeneration and a null allele is lethal (*Chu et al., 2009*).

Budding yeast Cdc48 (the human protein is known as p97 or valosin-containing protein [VCP]) is a member of the AAA (ATPases associated with diverse cellular activities) family of proteins. Cdc48 has intrinsic Ub-binding affinity that is greatly enhanced by the binding of UBX domain proteins or the Ufd1–Npl4 heterodimer to its N-terminus and Ufd2 to its C-terminus (*Ye, 2006*; *Meyer et al., 2012*). Cdc48/p97 is widely believed to function as a 'segregase' that uses the free energy released upon ATP hydrolysis to remodel ubiquitinated substrate complexes, usually, but not always, as a prelude to their degradation by the 26S proteasome (*Jentsch and Rumpf, 2007*). In the work reported here, we show that Cdc48–Ufd1–Npl4 promotes the release of tRNA-linked ubiquitinated nascent chains as well as aberrant non-stop and prematurely terminated polypeptides from the ribosome. Our findings establish a new function for Cdc48/p97 in RAD.

## Results

In an effort to systematically elucidate cellular functions of Cdc48 and its adaptors, mutants were subjected to chemical profiling by determining growth in the presence of various drugs, including translation inhibitors. All mutants were insensitive to anisomycin, an inhibitor that blocks the peptidyl transferase reaction on ribosomes (not shown). However, low concentrations of hygromycin B and paromomycin inhibited growth of multiple *cdc48* mutants and the adaptor mutants *npl4-1*, *ufd1*, *ubx1Δ, and ubx2Δ* (*Figure 1—figure supplement 1A,B* and data not shown). Hygromycin B and paromomycin belong to the aminoglycoside class of antibiotics that bind close to the decoding center and affect translational fidelity (*Carter et al., 2000*; *Kim and Craig, 2005*). Deletion of *LTN1* leads to hygromycin-sensitivity and since Ltn1 is involved in QC of NSD pathway substrates (*Bengtson and Joazeiro, 2010*), we explored the possibility that Cdc48–Ufd1–Npl4 and one or more Ubx proteins also function in this pathway.

To determine if Cdc48 contributes to protein degradation at the ribosome in unperturbed, exponentially growing cells, we purified ribosomes using a rapid, one-step affinity method designed to isolate 60S subunits, 80S monosomes and polysomes (*Oeffinger et al., 2007*; *Halbeisen and Gerber, 2009*; *Halbeisen et al., 2009*). Coomassie blue staining of purified ribosomes confirmed that Cdc48 activity was not required for their assembly (*Figure 1*, *Figure 1—figure supplement 2A*, lane 3), a conclusion that was further validated by sucrose gradient fractionation of ribosomes from wildtype, *cdc48-3*, and *ufd1-2* whole cell lysates (*Figure 1—figure supplement 2B*). However, the mutants do appear to have a slightly lower ratio of polysomes:80S monosomes. Immunoblotting for Ub revealed strong accumulation of Ub conjugates on ribosomes affinity-purified from *cdc48-3* cells (*Figure 1A*). To ascertain if the epitope tag on the ribosome subunit was exacerbating the effect of the *cdc48-3* mutation, we isolated ribosomes from untagged cells by pelleting through a sucrose cushion (*Halbeisen and Gerber, 2009*). Again, Ub conjugates were observed to accumulate on ribosomes in *cdc48-3* and *npl4-1* mutants (*Figure 1B*). Because the proteolytic defects of *npl4-1* are manifested at the semi-permissive temperature of 30°C (*Bays et al., 2001*), we carried out the experiment in *Figure 1B* at 30°C. The data indicate that *cdc48-3* does not have to be shifted to the fully non-permissive temperature to observe Ub conjugate accumulation at the ribosome. Additionally, conjugate accumulation on *cdc48-3* and *npl4-1* ribosomes exceeded that observed for ribosomes isolated from cells treated with the proteasome inhibitor MG132, a result that will be further discussed below.

To determine whether the ATPase activity of Cdc48 was required to prevent accumulation of ribosome-associated Ub conjugates, ribosomes were pelleted from *cdc48-3* mutant cells expressing plasmid-borne wildtype *CDC48*, or a mutant (Q2) deficient in ATPase activity (*Ye et al., 2003*). Whereas *CDC48* complemented the *cdc48-3* phenotype, high levels of Ub conjugates were detected on ribosomes isolated from cells expressing the *cdc48-Q2* mutant (*Figure 1C*). Consistent with the idea that Cdc48

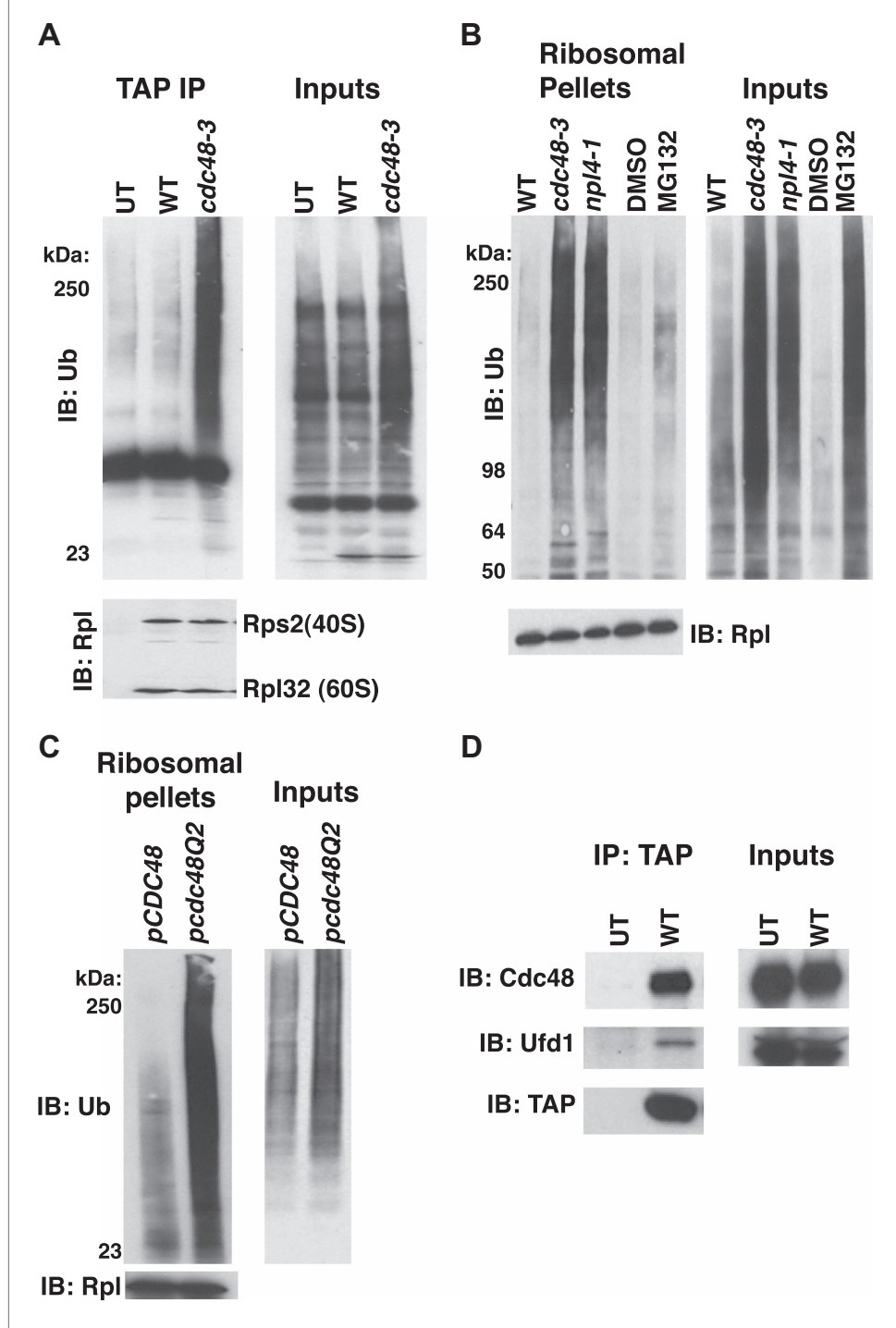

**Figure 1**. Ub conjugates accumulate on ribosomes isolated from *cdc48* and *npl4-1* mutants.(**A**) Ribosome assembly is unimpaired in *cdc48-3* mutant cells and Ub conjugates accumulate on *cdc48-3* ribosomes. Ribosomes were immunoprecipitated (IP) from *cdc48-3* (Untagged, UT), *RPL18BTAP* (WT), and *cdc48-3 RPL18BTAP* cells shifted to the non-permissive temperature (37°C) for 90 min. Purified ribosomes were evaluated by Coomassie blue staining (*Figure 1—figure supplement 2A*) and immunoblotting (IB) with Ub and anti-Rpl32 antibodies. Anti-Rpl32 also detects Rps2. (**B**) WT and mutant cells grown at 24°C were shifted to 30°C for two generations and *pdr5Δ* cells were either mock-treated with DMSO, or 30 μM MG132 for 30 min. Ribosomes were isolated (input cell lysates, right panel) by sedimentation through sucrose cushions. Ribosome pellets were evaluated by IB with Ub and Rpl32 antibodies (left panels). (**C**) The ATPase activity of Cdc48 promotes clearance of Ub conjugates from ribosomes.
*Figure 1. Continued on next page*

*Figure 1. Continued*

Mutant *cdc48-3* cells containing plasmid-borne WT *GAL-CDC48His6* or the Q2 mutant were grown in galactose for 2 hr to induce expression of ectopic Cdc48. Induced cells were shifted to 35°C for 1 hr to inactivate *cdc48-3* before being harvested. Ribosomes were isolated by sedimentation through sucrose cushions and input cell lysates and ribosome pellets were evaluated as in (**A**). (**D**) Cdc48 binds to purified ribosomes. Ribosomes were affinity-purified from untagged or *RPL18BTAP*-tagged cells and the levels of bound Cdc48 and Ufd1 were evaluated by IB with the respective antibodies.

The following figure supplements are available for figure 1:

**Figure supplement 1**. Cdc48 pathway mutants are sensitive to hygromycin B (**A**) and paromomycin (**B**).

**Figure supplement 2**. Ribosome assembly is unimpaired in Cdc48 pathway mutants.

ATPase functions at the ribosome to mediate release of ubiquitinated nascent peptides, both Cdc48 and Ufd1 were found to cofractionate with affinity-purified ribosomes isolated under stringent conditions (*Figure 1D*).

We next wished to ascertain if the Ub conjugates were derived from nascent polypeptides that were linked to tRNA. Magnetic beads containing bound ribosomes from wildtype or *cdc48-3* cells were treated with TEV protease, which cleaves between Rpl18B and the TAP tag. The TEV cleavage was carried out in the presence or absence of puromycin. Puromycin causes premature termination of translation by binding in the A-site of the ribosome and reacting with peptidyl-tRNA in the P-site, yielding a peptide-puromycin conjugate (*Nathans, 1964*). Thus, we expected that puromycin would induce the release of peptidyl-tRNA from the ribosome as a peptide-puromycin conjugate. The supernatants were then bound to UBA Sepharose resin to enrich for Ub conjugates. The rationale behind this step was that ribosome-associated Ub conjugates that remained bound to the ribosome would bind poorly to UBA beads because ribosomes do not efficiently penetrate the pores of a Sepharose resin, whereas Ub conjugates released from the ribosome by puromycin would be able to bind more efficiently to UBA beads. Puromycin treatment of ribosomes from *cdc48-3* but not wildtype cells released ubiquitinated proteins that were enriched on the UBA resin (*Figure 2A*, top panel). A similar result was obtained with ribosomes isolated from *npl4-1* and *ufd1-1* mutants (*Figure 2—figure supplement 1A,B*). The conjugates observed in *Figure 2A* were indeed linked to puromycin because they cross-reacted with an antibody specific for puromycin (*Figure 2A*, bottom panel). The same was true for Ub conjugates released from *npl4-1* and *ufd1-2* ribosomes (*Figure 2—figure supplement 1C*). The fact that no Ub immunoreactivity was recovered from mutant ribosomes in the absence of puromycin treatment validated our assumption that the UBA resin largely excluded mega-complexes. This experiment also suggests that at least some of the Ub conjugates that accumulated on *cdc48* and *npl4* ribosomes (*Figure 1A,B*) must be ubiquitinated nascent proteins linked to tRNA.

To further confirm that the Ub conjugates that accumulated on ribosomes in *cdc48* cells were ubiquitinated nascent peptides linked to tRNA we did three additional experiments. First, ribosomes affinity-purified from *cdc48-3* were treated with RNAse A or not before incubation with puromycin followed by binding to the UBA resin. Immunoblotting for Ub revealed that both the control and RNAse-treated ribosomes contained Ub conjugates that bound the UBA beads (*Figure 2B*, top panel, lanes 2 and 3). However, ribosomes pre-treated with RNAse no longer yielded puromycin-reactive conjugates (*Figure 2B*, bottom panel). By contrast, pre-treating lysate with RNAse *before* affinity purification of ribosomes eliminated recovery of both Ub-, and puromycin immunoreactivity (*Figure 2B*, lane 4, top and bottom panels).

The second experiment that we did to confirm tRNA linkage was to treat affinity-purified ribosomes with CTAB. CTAB is an ionic detergent that disrupts protein structure and precipitates peptidyl-tRNA molecules selectively (*Hobden and Cundliffe, 1978*; *Mariappan et al., 2010*). In parallel with results obtained previously using puromycin, Ub conjugates were precipitated from *cdc48-3* ribosomes in an RNAse-sensitive manner (*Figure 2C*).

As a final test of the idea that the Ub conjugates that accumulated on *cdc48-3* ribosomes were attached to nascent chains, wildtype and mutant cells were pulsed with radioactive methionine for 90 s and then chased with cold methionine. Ribosomes were affinity-purified from cells sampled at different times of chase and treated with puromycin to release tRNA-linked nascent chains. This material was

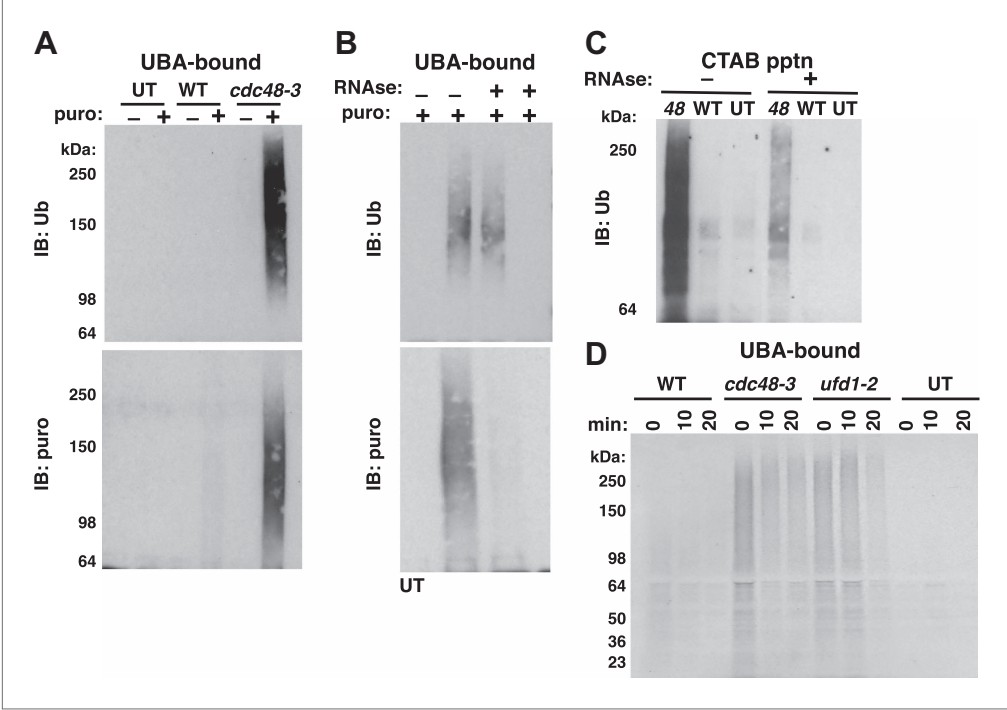

**Figure 2**. Ubiquitinated nascent peptides linked to tRNA accumulate on ribosomes in *cdc48-3* and *ufd1-2* mutants.(**A**) Puromycin-dependent binding to UBA resin of Ub conjugates from *cdc48-3* ribosomes. Ribosomes were affinity-purified from untagged or wildtype and mutant *RPL18BTAP* cells grown at 37°C for 90 min and eluted with TEV protease in the presence or absence of puromycin (puro), followed by incubation with UBA resin. Bound fractions were resolved by SDS-PAGE and immunoblotted with antibodies to Ub (top panel) or puromycin (lower panel). (**B**) Transfer of Ub conjugates to puromycin was RNAse A-sensitive. Ribosomes were affinity-purified from *cdc48-3* (UT) or *cdc48-3RPL18BTAP* cells as described above in the absence (lanes 1, 2 and 3) or presence of 200 μg/ml RNAse A. Following elution with TEV protease, samples from the tagged cells (lane 3) were treated with 200 μg/ml RNAse A at 30°C for 10 min before incubation of all samples with puromycin and binding to UBA resin. The bound fractions were evaluated as in (**A**). (**C**) CTAB-precipitable Ub conjugates accumulate on *cdc48-3* ribosomes. Ribosomes affinity-purified from the same strains used in panel (**A**) were treated with RNAse A (or not) and subjected to precipitation (pptn) with CTAB. Precipitates were resolved by SDS-PAGE and immunoblotted with anti-Ub. *48* Refers to *cdc48-3*. (**D**) Ubiquitinated newly-synthesized proteins accumulate on ribosomes isolated from *cdc48-3* and *ufd1-2* mutants. Cells were pulse-labeled for 90 s with $^{35}$S methionine and chased with cold methionine and cycloheximide for the indicated times. Ribosomes were affinity-purified, eluted with TEV protease in the presence of puromycin, and loaded onto UBA resin. The bound fraction was evaluated by SDS-PAGE followed by autoradiography. Densitometry indicated that <5% of the high MW material was released from *ufd1-2* ribosomes between the 0- and 10-min time points.

The following figure supplements are available for figure 2:

**Figure supplement 1**. Cdc48-Ufd1-Npl4 complex is required to clear ubiquitinated nascent peptides from ribosomes.

then fractionated on UBA resin to enrich for discharged Ub conjugates. Strikingly, ribosomes from both *cdc48-3* and *ufd1-2* cells contained far higher levels of radiolabeled high MW Ub conjugates than wildtype ribosomes, and a substantial portion of this signal persisted on ribosomes for 10–20 min (***Figure 2D***), which is considerably longer than the time required to synthesize a protein. Importantly, the low signal on wildtype ribosomes was not due to inefficient incorporation, because the WT inputs for the UBA fractionation step actually contained slightly more total pulse-labeled species (***Figure 2—figure supplement 1D***). Taken together, the experiments described thus far indicate that Cdc48 and its cofactors Ufd1 and Npl4 prevent accumulation of ubiquitinated, tRNA-linked nascent chains on the ribosome. For simplicity, we hereafter refer to these species as tUNPs (t̲RNA-linked, u̲biquitinated n̲ascent p̲olypeptides). Notably, other Cdc48 cofactors including Ubx2, Ubx4, Ubx6, and Ufd2 did not appear to contribute to clearance of tUNPs from the ribosome (***Figure 2—figure supplement 1C***).

These cofactors were tested because they had scored positively in the hygromycin-sensitivity assay whereas *ubx3Δ*, *ubx5Δ*, and *ubx7Δ* were hygromycin-insensitive, and therefore not pursued further. Hygromycin-sensitive *ubx1Δ/shp1Δ* was also not pursued further, because this mutant is extremely sick.

The findings summarized above raised the question: what is the composition of the tUNPs? One possibility is a class of aberrant proteins arising from translation of mRNAs that lack stop codons ('non-stop mRNA'). Translation through the poly(A) tail of these mRNAs can result in a ribosome that stalls either because it reaches the end of the message, or the polylysine encoded by the poly(A) tail inter-acts with the negatively charged exit tunnel of the ribosome (*Dimitrova et al., 2009*). Prior work has demonstrated that the Ub ligase Ltn1 targets proteins translated from mRNAs that lack a stop codon or that contain polylysine stretches (*Bengtson and Joazeiro, 2010*). We therefore employed the puromycin release/UBA bead capture protocol to test whether deletion of *LTN1* diminished the accumulation of tUNPs on ribosomes in *cdc48-3* cells. We also tested the impact of *ubr1Δ*, because of prior reports that Ubr1 promotes co-translational ubiquitination (*Turner and Varshavsky, 2000*), and because *ubr1Δ* mutants, like *ltn1Δ*, are hygromycin-sensitive (not shown, but see below). The data in *Figure 3A* demonstrate that additional loss of *LTN1* or *UBR1* resulted in a marked decrease in the level of tUNPs on *cdc48-3* ribosomes. This result encouraged us to evaluate the steady state levels of the Gfp-FLAG-His3Non-Stop (GFH$^{NS}$) fusion protein shown earlier to be an Ltn1 substrate at the ribosome that is degraded by the proteasome (*Bengtson and Joazeiro, 2010*). Amongst the Cdc48 pathway mutants examined, GFH$^{NS}$ accumulated maximally in *ufd1* mutants (similar to levels observed for *ltn1Δ*), and also exhibited significant accumulation in *npl4-1* and *cdc48-3* (*Figure 3B*). Cycloheximide chase analysis in *ufd1-2* cells indicated that accumulation resulted from stabilization of the non-stop reporter (*Figure 3C*). As observed in earlier studies (*Bengtson and Joazeiro, 2010*), the same reporter containing a stop codon was not degraded and accumulated at a much higher level than the NS reporter (*Figure 3C*; amounts loaded were adjusted to yield signals of equal intensity at 0 min).

If the Cdc48–Ufd1–Npl4 complex functions downstream of Ltn1 to mediate clearance of tUNPs from the ribosome, as suggested by the data in *Figure 3A*, we anticipated that ubiquitinated species of the non-stop reporter should accumulate upon inactivation of Cdc48 complex. To address this question, we prepared SDS lysates of wildtype and *ufd1* mutant cells expressing GFH$^{NS}$, diluted the lysates with Triton X-100 to sequester the SDS in micelles, and immunoprecipitated with anti-FLAG. Immunoblotting for Ub demonstrated that GFH$^{NS}$ accumulated in an ubiquitinated form in *ufd1-1* and *ufd1-2* but not in *ltn1Δ* mutants (*Figure 3D*).

The above data indicate that the non-stop, but not the stop codon-containing reporter was targeted for degradation by the combined actions of Ltn1 and Cdc48–Ufd1–Npl4. The role of the Cdc48 complex in reporter degradation appeared to be direct, because both Cdc48 and Ufd1 were co-immunoprecipitated with GFH$^{NS}$, but not GFH$^{Stop}$, even though GFH$^{Stop}$ was more abundant (*Figure 3E*).

GFH$^{NS}$ accumulates on 80S ribosomes in *ltn1Δ* mutants (*Bengtson and Joazeiro, 2010*). To evaluate if any of these molecules are linked to tRNA, we sedimented ribosomes rapidly through sucrose cushions and treated the pellet with RNAse A (or not). Strikingly, immunoblotting revealed an RNAse-sensitive species on ribosomes from *ltn1Δ*, as well as those from *cdc48-3* and *ufd1-2* mutants, but not from wildtype cells (*Figure 3—figure supplement 1*). The fraction of GFH$^{NS}$ linked to tRNA was low. It is not clear whether this was due to hydrolysis of the tRNA linkage in vivo or in vitro.

For our subsequent experiments, we decided to focus on Protein A-NS (PrA$^{NS}$), a non-stop reporter utilized in earlier studies (*Wilson et al., 2007*; *Bengtson and Joazeiro, 2010*). Unlike GFH$^{NS}$, the RNAse-sensitive, tRNA-linked form of PrA$^{NS}$ accumulated more efficiently and could be detected even in total cell extracts. We first examined the steady-state levels of PrA$^{NS}$ in several mutants. Interestingly, levels of tRNA-linked PrA$^{NS}$ in *cdc48-3* and *ltn1Δ* cell lysates were equivalent to those detected in *dom34Δ* and *ski7Δ*, bona fide components of the NSD pathway (*Figure 4A*; equivalent loading confirmed by staining with Ponceau S and immunoblotting for tubulin, *Figure 4—figure supplement 1A*).

To more precisely determine which subset of ribosomes contained tRNA-linked PrA$^{NS}$, we performed sucrose gradient fractionation. Shallow gradients (10–30%) were set up, enabling good resolution of the subunits, monosomes, and polysomes (*Figure 4—figure supplement 1B*). Immunoblotting for PrA$^{NS}$ revealed that the RNAse-sensitive species peaked in the 80S fractions isolated from *ltn1Δ* mutants but was found in both the 60S and 80S-containing fractions from *cdc48-3* cells (*Figure 4B*). Additionally, modified forms of slower mobility than the tRNA-linked species were detected in 60S ribosomes from *cdc48-3* but not *ltn1Δ*, suggesting that these high MW species are Ltn1-dependent Ub conjugates. Only a small fraction of the PrA$^{NS}$ detected on ribosomes migrated as Ub-conjugated

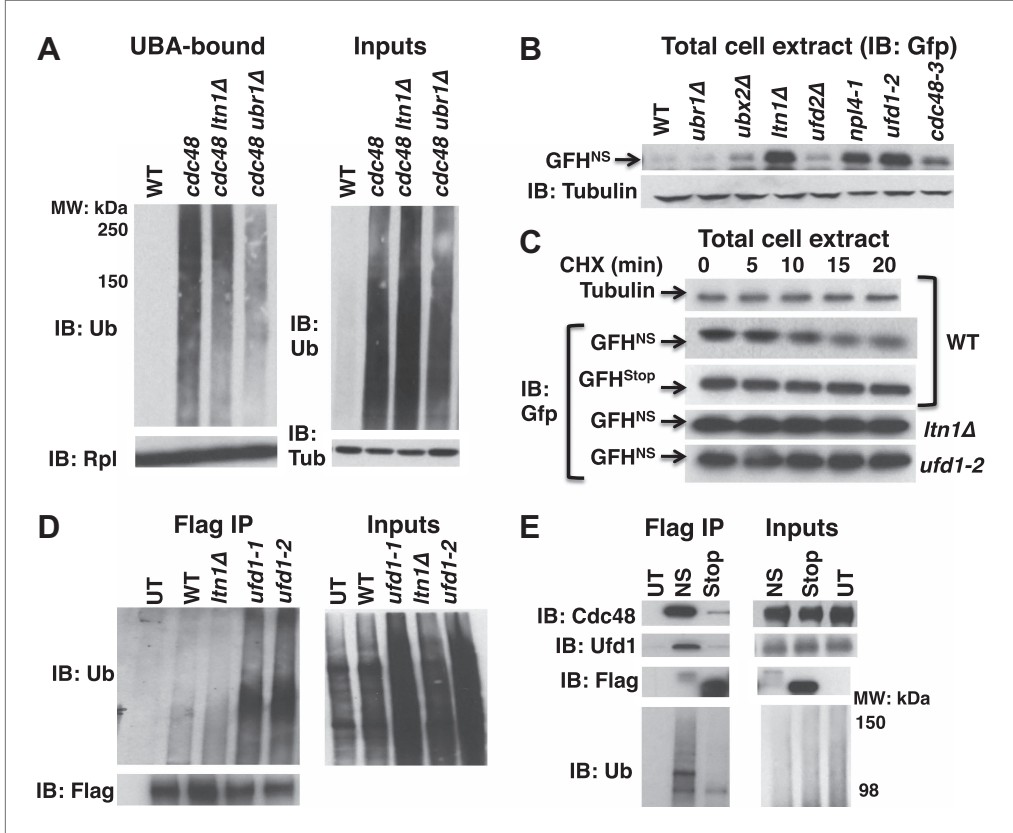

**Figure 3**. Aberrant proteins derived from non-stop mRNA accumulate in cells deficient in Cdc48–Ufd1–Npl4 function.(**A**) Ltn1 and Ubr1 contribute to accumulation of Ub conjugates on *cdc48-3* ribosomes. Ribosomes from the indicated strains grown at 30°C were isolated from input cell lysates (right panels) by pelleting through sucrose cushions, treated with puromycin, and incubated with UBA resin. The bound fraction (left panels) and inputs were evaluated by SDS-PAGE and immunoblotting with Ub, tubulin (Tub), and Rpl32 antibodies as indicated. The *-3* allele of *cdc48* was used. (**B**) The non-stop reporter GFH[NS] accumulates in Cdc48 pathway mutants. Glass bead/SDS extracts from exponential cultures (grown at 30°C) of the indicated mutants harboring a plasmid that expresses GFH[NS] were analyzed by SDS-PAGE and IB with anti-Gfp. Tubulin served as the loading control. (**C**) Cycloheximide chase analysis of cells expressing either the non-stop (GFH[NS]) or stop codon-containing (GFH[Stop]) reporters. Glass bead/SDS extracts prepared from aliquots harvested at the indicated times from wildtype and mutant cultures were analyzed by SDS-PAGE and IB with anti-Gfp. Samples from WT expressing GFH[NS] were also evaluated by IB with anti-tubulin to confirm equal loading. Note that extracts prepared from cells expressing GFH[Stop] were loaded at one-fifth the amount of GFH[NS]. (**D**) Non-stop protein accumulates in the ubiquitinated state in *ufd1* mutants. Glass bead/SDS extracts of wildtype and mutant cells expressing plasmid-borne GFH[NS] and grown at 37°C for 90 min were evaluated directly (inputs) or immunoprecipitated with anti-Flag antibodies after 10-fold dilution with buffer containing Triton X-100. Bound proteins and inputs were evaluated by SDS-PAGE and IB with anti-Ub and Flag antibodies. UT corresponds to WT cells not expressing GFH[NS]. (**E**) Cdc48 and Ufd1 interact selectively with non-stop protein. Lysates prepared from cells expressing Flag-tagged GFH[NS] (NS) or GFH[Stop] (Stop) reporters as well as cells lacking a plasmid (UT) were immunoprecipitated with anti-Flag antibodies. Total cell extracts (inputs; right panels) and bound proteins (left panels) were evaluated by SDS-PAGE and IB with anti-Cdc48, Ufd1, Flag and Ub antibodies.

The following figure supplements are available for figure 3:

**Figure supplement 1**. Ribosomes from wildtype and mutant cells grown at 37°C for 90 min were sedimented through sucrose cushions.

species. This is consistent with what is typically observed for most Ub-dependent substrates upon inhibition of their degradation, and presumably reflects robust cytosolic deubiquitination activity. In addition, overproduction of PrA[NS] from the *GAL* promoter may overwhelm Ltn1, which is inabundant (***Bengtson and Joazeiro, 2010***). Besides the modified forms of PrA[NS], unmodified protein was observed

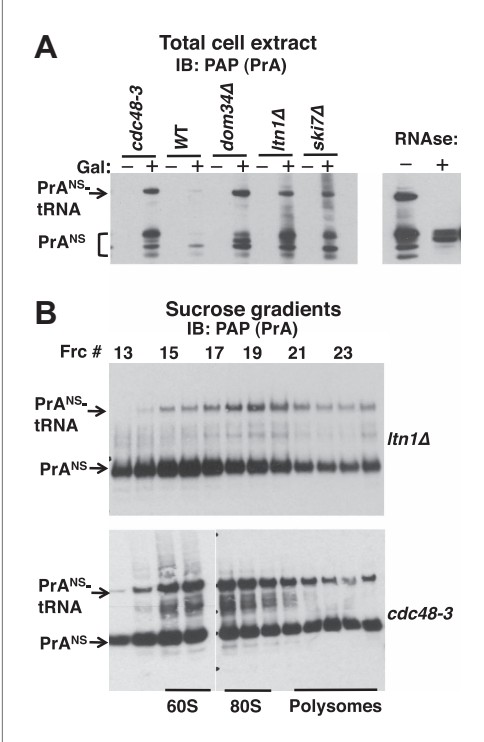

**Figure 4**. Non-stop reporter PrA<sup>NS</sup> accumulates on 60S and 80S ribosomes in a tRNA-linked form in *cdc48-3* cells.(**A**) NuPAGE gel analysis reveals accumulation of tRNA-linked PrA<sup>NS</sup> in *cdc48-3* and *ltn1Δ* cells. Wildtype and mutant cells containing reporter plasmid were grown at 30°C for one generation and expression of reporter was induced (or not) by the addition of 2% galactose (±Gal) for 2 hr. Glass bead/SDS extracts were fractionated on a NuPAGE gel to preserve tRNA-linked species, and analyzed by IB with PAP to detect protein A. Equal loading was confirmed by Ponceau S staining and IB for tubulin (**Figure 4—figure supplement 1A**). The right panel shows collapse of the tRNA-linked species in *cdc48-3* extract following RNAse treatment. (**B**) tRNA-linked PrA<sup>NS</sup> accumulates on 60S and 80S ribosomes in *ltn1Δ* and *cdc48-3* mutants. Sucrose gradient (10–30%) fractions from cells expressing plasmid-borne PrA<sup>NS</sup> and grown at 30°C for two generations were concentrated by TCA precipitation and evaluated by fractionation on a NuPAGE gel and IB with anti-PAP or anti-Rpl32 (**Figure 4—figure supplement 1B**). Fractions 15 and 16 (enriched for 60S) and 17–19 (enriched for 80S ribosomes) are indicated.

The following figure supplements are available for figure 4:

**Figure supplement 1**. Fractionation of ribosome subunits on sucrose gradients.
DOI: 10.7554/eLife.00308.0011

throughout the gradient and peaked in the ribosomal fractions. We do not know whether this latter form arose from hydrolysis of the peptidyl-tRNA linkage in cells or in vitro.

Our data thus far indicate that in *cdc48, npl4,* and *ufd1* mutants, nascent proteins were targeted for degradation by Ltn1- and Ubr1-dependent ubiquitination but became stuck on 60S and 80S ribosomes as peptidyl-tRNAs, implying that Cdc48 activity promotes release of stalled, ubiquitinated peptides, which are then degraded by the proteasome. This hypothesis predicts that ribosomes isolated from cells in which proteasome activity is compromised should accumulate less tUNPs than *cdc48* mutants. To test this prediction, we used the sucrose cushion method to isolate ribosomes from *cdc48-3* and *npl4-1* mutants, as well as *pdr5Δ* cells treated with or without the proteasome inhibitor MG132 (*pdr5Δ* enables cellular accumulation of this compound). The ribosome pellet was then treated with puromycin and incubated with UBA resin to enrich for tUNPs. Ribosomes isolated from cells with inhibited proteasomes contained far less tUNPs than ribosomes from *cdc48-3* and *npl4-1* cells (**Figure 5A**; the input lysates and ribosome pellets for this experiment are shown in **Figure 1B**). This result is in striking contrast to what we previously reported for proteasomes isolated from *cdc48-3* and MG132-treated cells—*both* contained very high levels of accumulated Ub conjugates (**Verma et al., 2011**). A similar result was obtained using the *cim3-1* mutant instead of MG132 treatment (**Figure 5—figure supplement 1A**; *cim3-1* is a ts allele of the gene that encodes the proteasome subunit Rpt6).

We next sought to evaluate the effect of proteasome deficiency on accumulation of a specific peptidyl-tRNA on the ribosome. To evaluate the subcellular distribution of the PrA<sup>NS</sup> species that accumulate upon inhibition of Cdc48 or proteasome function, we sedimented extracts through sucrose cushions to generate ribosome pellets and post-ribosomal supernatants from wildtype, *cdc48-3* and *cim3-1* cells (**Figure 5B**). Notably, whereas most of the PrA<sup>NS</sup> fractionated in the ribosome pellet in *cdc48-3* cells (a substantial portion of which remained linked to tRNA), the opposite was true in *cim3-1* cells, where the bulk of the PrA<sup>NS</sup> was recovered in the supernatant fraction (**Figure 5B**). Comparison of extracts made by lysing cells directly in SDS (**Figure 5—figure supplement 1B**) or under native conditions (**Figure 5B**) revealed that the proportion of PrA<sup>NS</sup> that accumulated as full length or truncated species in vivo and its susceptibility to proteolytic clipping in native cell extract differed for *cdc48-3* and *cim3-1* mutants. We do not know the basis for these differences.

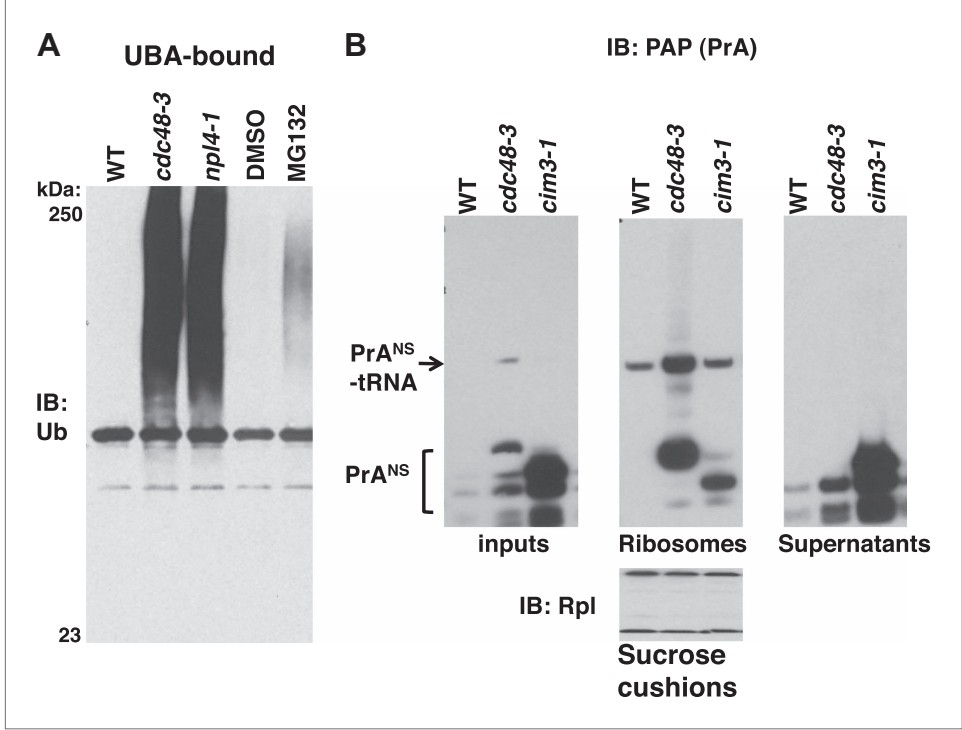

**Figure 5**. Non-stop reporter PrA[NS] accumulates on ribosomes in *cdc48-3* cells but is released from ribosomes in cells deficient in proteasome activity.(**A**) Ub conjugates accumulate to a greater extent on ribosomes isolated from *cdc48-3* and *npl4-1* mutants than on ribosomes isolated from cells treated with proteasome inhibitor MG132. Puromycin-treated ribosomes from *Figure 1B* were incubated with UBA resin. The bound fraction was evaluated by SDS-PAGE and immunoblotting with anti-Ub. (**B**) PrA[NS] preferentially accumulates on the ribosome in *cdc48-3* cells and in the post-ribosome supernatant in a proteasome mutant. Ribosomes were isolated from cells grown at 37°C for 90 min by pelleting through sucrose cushions. Total cell lysate inputs (left panel), ribosome pellets (middle panel) and post-ribosome supernatants (right panel) were evaluated by separation on a NuPAGE gel and IB with PAP. Equivalent recovery of ribosomes in the pellet fractions was confirmed by IB with anti-Rpl32 (bottom panel). The same cell cultures were also evaluated by preparing extracts in SDS and resolving aliquots on a Tris-Glycine (*Figure 5—figure supplement 1B*, left panel) or NuPAGE (*Figure 5—figure supplement 1B*, right panel) gel.

The following figure supplements are available for figure 5:

**Figure supplement 1**. Maximal Ub conjugate accumulation on ribosomes isolated from *cdc48-3* mutant cells.

Our observation that Ubr1 contributed to ubiquitination of tUNPs (*Figure 3A*) led us to wonder about the diversity of the QC mechanisms that underlie their formation. To determine if the role of Cdc48 is limited to the non-stop pathway or whether it plays a more global role in degradation of proteins encoded by aberrant messages, we also assessed the stability of endogenous phosphogluconate dehydrogenase truncated by a premature termination codon (Gnd1[PTC]) in the middle of a folded domain at residue 368. Over-produced Gnd1[PTC] was shown to be a substrate of the Ubr1 cytosolic quality control pathway in a prior study (*Heck et al., 2010*). The data in *Figure 6A* confirmed stabilization of Gnd1[PTC] in *ubr1Δ*. Notably, *cdc48-3* and *ufd1-2* mutations also stabilized Gnd1[PTC], but *ltn1Δ* was without effect (equivalent loading confirmed in *Figure 6—figure supplement 1*). Gnd1[PTC] was also stabilized in *upf1Δ*. Upf1 is a RING domain Ub ligase (*Takahashi et al., 2008*) that plays a crucial role in NMD and stimulates the degradation of aberrant PTC translation products (*Kuroha et al., 2009*). The dependence of Gnd1[PTC] degradation on Cdc48 and Upf1 suggested that tRNA-linked Gnd1[PTC] might become stuck on ribosomes, as observed for the non-stop reporters. To test this possibility we used the sucrose cushion method to isolate ribosomes from various mutants, and evaluated the ribosomal pellets for their content of Gnd1[PTC]. Analysis of total cell lysates revealed strong accumulation of Gnd1[PTC] in *cdc48-3*, *ubr1Δ*, and *upf1Δ* cells (*Figure 6B*). Fractionation of lysates prepared in an identical manner revealed an RNAse-sensitive species on ribosomes but not in post-ribosomal supernatants

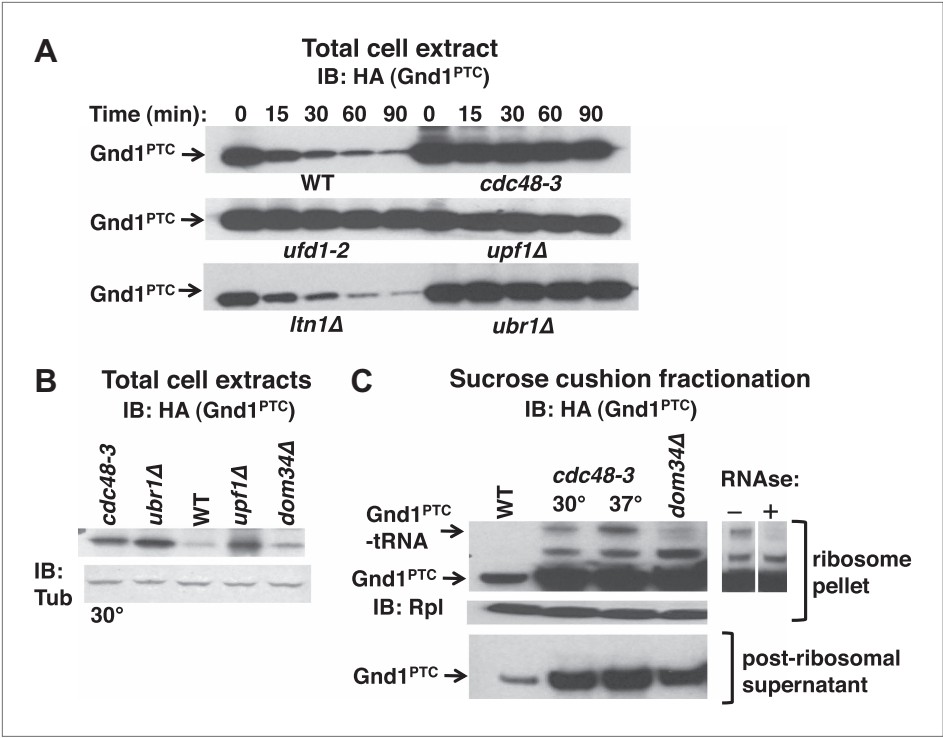

**Figure 6**. The Cdc48 pathway is required for degradation of prematurely terminated Gnd1$^{PTC}$. (**A**) A cycloheximide chase was performed with the indicated mutants grown at 30°C for two generations. Glass bead/SDS extracts were prepared at the indicated times after adding cycloheximide and analyzed by immunoblotting with anti-HA antibody to detect HA-tagged Gnd1$^{PTC}$. (**B**) A tRNA-linked form of Gnd1$^{PTC}$ accumulates on ribosomes from *cdc48-3* mutants. Total cell extracts prepared identically to those used for the ribosome analysis in panel C were immunob-lotted with anti-HA to detect Gnd1$^{PTC}$ and anti-tubulin to evaluate loading. *cdc48-3* mutant cells were incubated at a semi-permissive (30°C) temperature for 90 min prior to lysis whereas all other strains were grown at 30°C for two generations. (**C**) Ribosomes (top panels) and post-ribosomal supernatants (bottom panel) were isolated from the same strains in (**B**) by sedimentation through sucrose cushions and analyzed by immunoblotting with anti-HA to detect Gnd1$^{PTC}$. The ribosome pellets were also immunoblotted with anti-Rpl32 (middle panel) to confirm equiva-lent recovery. The right panel shows collapse of the tRNA-linked species following RNAse treatment of the ribosomal pellet from *cdc48-3*. *cdc48-3* cells were incubated at either the semi-permissive (30°C) or non-permissive (37°C) temperature prior to lysis, as described in (**B**).

The following figure supplements are available for figure 6:

**Figure supplement 1**. Ponceau S staining of filters for **Figure 6A**.

isolated from *cdc48-3* mutants (**Figure 6C**) as well as *ufd1-2*, *ubr1Δ* and *upf1Δ* mutants (data not shown). In contrast, little tRNA-linked substrate was found in *dom34Δ* mutants (**Figure 6C**). Strikingly, the tRNA-linked species was not detected on ribosomes from *ltn1Δ* (data not shown).

## Discussion

Whereas QC of defective mRNAs and recycling of ribosomes stalled on these templates has received considerable attention in prior studies, much less is known about what happens to the nascent polypeptides, some of which accumulate as peptidyl-tRNA. Some proteins encoded by non-stop mRNAs, including the reporter proteins used here (GFH$^{NS}$ and PrA$^{NS}$), are degraded by the ubiquitin-proteasome system (UPS) (**Wilson et al., 2007**; **Bengtson and Joazeiro, 2010**). The Ub ligase Ltn1 binds 60S ribosomes and promotes ubiquitination of stalled GFH$^{NS}$ (**Bengtson and Joazeiro, 2010**). Our data suggest that Cdc48–Ufd1–Npl4 acts downstream of Ltn1 to promote release of the non-stop peptide from the ribosome, so that it can be degraded by the proteasome. How Ltn1 and the Cdc48 complex does this remains unknown. Indeed, this remains unknown for any NSD or NGD pathway

substrate; the Dom34-Hbs1 complex lacks the residues necessary for hydrolysis of the peptidyl-tRNA bond, and the two budding yeast peptidyl-tRNA hydrolases, Pth1 and Pth2, are non-essential mitochondrial proteins whose transmembrane configurations are currently unknown (**Rosas-Sandoval et al., 2002**). An interesting question for the future will be to determine whether Ltn1 and Cdc48–Ufd1–Npl4 act together with, or in a redundant pathway parallel to Ski7 or Dom34–Hbs1-Rli1.

We observed direct physical association of Cdc48 and Ufd1 with both affinity-purified ribosomes and the NSD pathway substrate GFH$^{NS}$. This interaction was highly specific, because Cdc48 and Ufd1 did not bind the same substrate when it was translated from a message that includes a stop codon. While this manuscript was in revision, it was reported that Cdc48, Ufd1, and Npl4 are components of the newly identified Ribosome Quality Control (RQC) complex, which also contains Ltn1, Rqc1 and Tae2 and associates with the 60S ribosomal subunit. The latter two proteins, like Ltn1 and Cdc48–Ufd1–Npl4, are highly conserved and have mammalian homologs (**Brandman et al., 2012**). Both Ltn1 and Rqc1 help link Cdc48–Ufd1–Npl4 to RQC. Rqc1 associates with and is required for the degradation of an Ltn1 substrate that contains an internal polyarginine sequence, but a functional role for Cdc48–Ufd1–Npl4 in RQC was not established by **Brandman et al. (2012)**.

A key question that emerges from this work is, what is the signal that differentiates ribosomes engaged with NSD or other aberrant substrates from normally translating ribosomes? Ltn1 (**Bengtson and Joazeiro, 2010**) and RQC (**Brandman et al., 2012**) are reported to be predominantly bound to 60S ribosomal subunits, suggesting that they may act downstream of the initial recognition and splitting of a stalled ribosome. However, in *ltn1Δ* cells, GFH$^{NS}$ accumulates on 80S monosomes (**Bengtson and Joazeiro, 2010**). Likewise, we report here that tRNA-linked PrA$^{NS}$ appeared to be concentrated in monosomes in *ltn1Δ*. By contrast, tRNA-linked PrA$^{NS}$ was spread more broadly across 60S and 80S fractions in extracts from *cdc48-3* cells. The different fractionation behavior of tRNA-linked PrA$^{NS}$ in *ltn1Δ* and *cdc48* mutants may reflect leakiness of the *cdc48* mutation (the experiment was done at the semi-permissive temperature of 30°C) or Cdc48-independent ribosome splitting. Detailed resolution of the exact sequence of events will require reconstitution of the overall process with defined components.

In addition to observing accumulation of engineered NSD pathway reporter proteins in Cdc48 pathway mutants, we also observed extensive accumulation of endogenous, tRNA-linked ubiquitinated nascent polypeptides (tUNPs) on ribosomes in these cells. These tUNPs could potentially arise from multiple sources including misfolded domains (**Turner and Varshavsky, 2000**), mRNAs whose transcription was prematurely terminated, mRNAs or nascent peptides containing sequences that cause ribosome pausing (**Ingolia et al., 2011**) or mRNAs that encode premature termination codons (e.g., **Figure 6C**). Thus it is perhaps not surprising that at least two Ub ligases (Ltn1 and Ubr1) contributed to their accumulation. In any event, we believe it is unlikely that there is a single pathway or mechanism for disposing of tUNPs. Even in the seemingly simple case of a ribosome that translates into the poly(A) tail of an mRNA that is prematurely polyadenylated within the coding sequence, the 3′-most ribosome may be stalled because it reached the end of the message, or because polylysine encoded by the poly(A) tail bound the exit channel of the ribosome (**Ito-Harashima et al., 2007**). Because stalls in translation elongation lead to endonucleolytic cleavage of the mRNA upstream of the 3′-most ribosome (**Tsuboi et al., 2012**), each trailing ribosome will present a unique substrate to the UPS, and may be processed by a different mechanism.

The intimacy with which protein biosynthesis is coupled to protein degradation is highlighted by the identification of the translasome, a supercomplex defined by mass spectrometric analysis of the translation initiation factor eIF3 and found to contain both ribosome and proteasome subunits as well as chaperones involved in protein QC (**Sha et al., 2009**). A tight coupling of these processes may be particularly important for neurons, which are post-mitotic and need to remain robust for the life of the organism (**Segref and Hoppe, 2009**). Indeed, homozygous lister mice deficient in Listerin (Ltn1) function exhibit severe neurological dysfunction (**Chu et al., 2009**), whereas mutations in human Ubr1 result in the Johanson–Blizzard syndrome characterized by, amongst other phenotypes, frequent mental retardation (**Hwang et al., 2011**). Mutations in p97 contribute to 1–2% of familial ALS (**Johnson et al., 2010**) and are the root cause of a syndrome, IBMPFD, that includes frontotemporal dementia (**Watts et al., 2004**). Defects in metabolism of protein aggregates (**Manno et al., 2010**), endosomal trafficking (**Ritz et al., 2011**) and autophagy (**Ju et al., 2009**; **Tresse et al., 2010**) have been previously suggested to contribute to the pathophysiology that results from mutation of p97. As the molecular mechanisms underlying these diseases are unraveled, it will be interesting to see if ribosome-associated degradation contributes to their etiology.

## Materials and methods

### Yeast strains and growth

All strains used in this study are listed in *Supplementary file 1*. They were derived from the W303 or S288C backgrounds. Deletion strains were acquired from Open Biosystems (Waltham, MA). All crosses were confirmed by auxotrophic marker selection and genomic PCR. Temperature-sensitive strains were confirmed by incubating the plate at the restrictive temperature for 4–5 days. Cultures were grown in YPD, or if cells contained plasmids, in synthetic selection medium. For expression of PrA[NS] from the *GAL1,10* promoter, induction was typically for 2 hr. Temperature-sensitive mutants and their congenic wildtypes were grown at 25°C and temperature shifts were for 90 min at 37°C, or two cell division cycles at 30°C. Any exceptions to these regimens are noted in the respective figure legends.

### Preparation of glass bead/SDS extracts

Cells were grown as described in the respective figure legends to a final $O.D._{600}$ (optical density at a wavelength of 600 nm) between 1.0 and 2.0, harvested by centrifugation, and drop-frozen in liquid nitrogen. Frozen cell pellets were thawed and washed with ice-cold buffer containing 50 mM Tris, pH 7.5, 10 mM sodium azide, 10 mM EDTA, 10 mM EGTA, 1× protease inhibitor tablet (Roche, Basel, Switzerland), 10 mM NEM, 50 mM NaF, 60 mM β-glycerophosphate, 10 mM sodium pyrophoshate. An equal volume of glass beads (Sigma, St Louis, MO; 425–600 µm, acid washed) was added and the cell pellets were immersed in boiling water for 3 min. They were then suspended in 1× SDS buffer (37.5 µl/O.D. unit) and cells were lysed by vortexing in Fast Prep-24 (MP) for 45 s at a setting of 6.5, and boiled again for 4 min. Boiled lysates were centrifuged at 16,000×*g* for 1 min. Aliquots were resolved by SDS-PAGE. In some instances, particularly when tRNA-linked substrate was being detected, washed cell pellets were brought up in 1× SDS buffer containing 1× Secure (Ambion, Carlsbad, CA), heated at 65°C for 5 min, lysed by glass beads as described above, and loaded on NuPAGE gels (Invitrogen, Carlsbad, CA). For immunoblot detection, gels were transferred to nitrocellulose and filters were stained with Ponceau S to determine equivalent loading of protein extracts. If loading was not equivalent, the experiment was repeated. The nitrocellulose filters were immunoblotted with desired antibody and developed by ECL or Super Signal (Pierce, Rockford, IL). Anti-Gfp was from Clontech (Mountain View, CA), Peroxidase Anti-Peroxidase Soluble Complex (for all Protein A detections, PAP), anti-Tubulin and anti-Flag were from Sigma, anti-TAP was from Thermo (Waltham, MA), anti-HA (12CA5) was from Roche, and anti-Ub antibodies were from Chemicon (Millipore, Billerica, MA), Stressgen (Enzo Life Sciences, Farmingdale, NY), and Abcam (Cambridge, MA). Puromycin antibody was a gift from Peter Walter, and anti-Rpl32 a gift from Jonathan Warner.

### Affinity purification of ribosomes

Ribosomes were affinity-purified using TAP-tagged *RPL18B* by following a protocol established in prior studies with some modifications as follows (*Halbeisen and Gerber, 2009*; *Halbeisen et al., 2009*). Exponential cultures were grown at either 30°C or 37°C for 1.5 hr before being harvested by filtration using 0.2 µm Nalgene filters. In some experiments 0.1 mg/ml cycloheximide was added during filtration, and was omitted when no difference was observed in the experimental outcome. Cells were washed with a buffer containing 50 mM Tris, pH 7.5, 10 mM NEM, 10 mM sodium pyrophosphate, 0.1 mg/ml cycloheximide and frozen in liquid nitrogen. Pellets were ground in liquid nitrogen. Cell powder was weighed and resuspended in twice the volume of lysis buffer (buffer A) containing 20 mM Tris–HCl (pH 8.0), 140 mM KCl, 5 mM $MgCl_2$, 1% Triton-100, 10% glycerol, 0.2 mg/ml heparin, 0.1 mg/ml cycloheximide, 5 mM NEM, 0.5 mM AEBSF, protease inhibitor cocktail (EDTA-free; Roche), 20 U/ml DNAse I, 0.05 U/µl of RNaseOut (Invitrogen), 0.05 U/µl of SUPERase-In (Ambion), 0.5 mM DTT. Lysate was centrifuged at 5000 rpm for 5 min and the supernatant reclarified at 14,000 rpm for 10 min in a refrigerated microfuge (Eppendorf, Hamburg, Germany; 5417R). Lysate was bound to rabbit IgG conjugated to superparamagnetic Dynabeads (Invitrogen) as described by (*Oeffinger et al., 2007*). Using beads of this diameter enables the isolation of large complexes such as polysomes. Magnetic beads were washed twice with wash buffer (buffer B) containing 20 mM Tris–HCl, pH 8.0, 140 mM KCl, 5 mM $MgCl_2$, 10% glycerol, 0.025 U/µl RNaseOut, 0.025 U/µl of SUPERase-In, 1 mM DTT, 0.1% NP40, supplemented with 0.1 mg/ml cycloheximide and 5 mM NEM, and then twice with buffer B lacking supplements. Elution was in buffer B containing 0.3 U/µl of Tobacco Etch Virus (TEV) protease at 16°C for 2 hr with intermittent shaking in the presence (or absence) of 2 mM puromycin, following which samples were adjusted to 0.6 M KCl and 3 mM puromycin and incubated at 37°C for 7 min. Eluates were removed

after immobilization of beads with a magnet and characterized by Coomassie blue staining, immunoblotting, and RNA analysis using RNeasy mini columns (Qiagen, Valencia, CA).

## Sucrose density fractionation

A total of 25 $A_{260}$ units of cultures lysed in buffer A were loaded on top of a 10–50% sucrose gradient prepared in buffer A lacking Triton X-100, glycerol and NEM. The samples were centrifuged in an SW-41 rotor for 180 min at 35,000 rpm at 4°C and fractionated while continuously recording the absorbance at 254 nm with a flow cell UV detector (ISCO, Lincoln, NE). Fractions of 0.5 or 0.75 ml were collected and analyzed by Coomassie blue staining and immunoblotting with various antibodies, including Rpl32, to identify 40S, 60S, 80S and polysomal fractions. An aliquot (100 µl) of each fraction was also analyzed for RNA using RNeasy mini columns (Qiagen). *Figure 1—figure supplement 2B* is representative of this protocol.

Ribosomes were also analyzed on 10–30% gradients prepared in 50 mM Hepes, pH 7.4, 2.5 mM MgCl$_2$ and 100 mM KOAc using the SW55Ti rotor at 50,000 rpm for 100 min with slow acceleration / deceleration. 0.2 ml fractions were collected from the top. Aliquots were resolved by SDS-PAGE and immunoblotted with anti-Rpl32 and the Ab for the specific NS substrate. To aid in tRNA detection, samples were also precipitated with 10% TCA following the addition of 0.02% deoxycholate. Pellets were washed with ice-cold acetone, resuspended in 2× Laemmli buffer containing RNA secure and analyzed by NuPAGE. Representative data are shown in *Figure 4B*.

For analysis of tRNA-linked substrates in *Figures 5B and 6B* and *Figure 3—figure supplement 1*, we employed the method described by (*Tsuboi et al., 2012*). Ribosomes were sedimented at 49,000 rpm for 45 min (RP100AT4 rotor in Sorvall RC M120EX ultracentrifuge) through 0.5 M sucrose cushions in buffer C containing 20 mM Hepes, pH 7.4, 5 mM MgOAc, 100 mM KOAc, 0.5 mM DTT, 100 µg/ml cycloheximide, 200 µg/ml heparin, and the protease and RNAse inhibitor cocktails described above. Ribosome pellets were resuspended in the same buffer in the presence (or absence) of 0.5 M KCl, 3 mM puromycin, 2 µM Ub aldehyde and incubated for 15 min on ice and 30 min at 25°C. RNAse inhibitors were omitted when ribosomes were treated with 10 or 100 µg/ml RNAse A at 30°C for 10 min.

### Binding to UBA columns

Two types of UBA resins were used to bind multiubiquitinated substrates: purified and dialyzed recombinant Gst-Dsk2 immobilized on glutathione-sepharose beads (*Figures 2, 5A* and *Figure 2—figure supplement 1C*), TUBE2-UBA (Boston Biochem, Cambridge, MA; *Figure 2—figure supplement 1B*, and *Figure 3A*). Purified ribosomes in high salt and puromycin were diluted with buffer containing 20 mM Tris–HCl, pH 7.5, 5 mM NEM, 0.1% NP40 and bound to UBA resin at 4°C for 1.5 hr. Beads were washed three times with 25 mM Tris, pH 7.5, 150 mM NaCl, 0.2% NP40 and a final wash was performed with the same buffer minus detergent. Beads were re-suspended in 2× SDS-PAGE buffer and loaded on 4–12% Tris-Glycine gels (Invitrogen). Following transfer, they were typically immunoblotted with anti-Ub antibody.

### CTAB precipitation of peptidyl-tRNA

Samples to be precipitated were adjusted to 2% CTAB (Hexadecyltrimethyl-ammonium bromide; Sigma) and mixed with an equal volume of 0.5 M sodium acetate containing 0.2 mg/ml bacterial tRNA (*Hobden and Cundliffe, 1978*; *Mariappan et al., 2010*). Samples were incubated at 30°C for 10 min and then centrifuged for 10 min at room temperature. Precipitates were washed with 1% CTAB, 0.25 M sodium acetate at room temperature and resuspended in SDS-PAGE buffer.

## [$^{35}$S] labeling of cells

Wildtype and mutant cells growing in synthetic medium were temperature shifted to 36°C for 1 hr and starved for methionine for 30 min. Cells were then harvested by centrifugation, resuspended in one-tenth the original volume, and pulse-labeled with 150 µCi/ml $^{35}$S-methionine (>600 Ci/mmol, cell-labeling grade; Perkin Elmer, Waltham, MA) for 90 s. Aliquots were immediately withdrawn into collection tubes containing 2× stop solution (0.2 mg/ml cycloheximide and 500 mM sodium azide), and the remainder of the cultures chased for the indicated times with 0.2 mg/ml cycloheximide and 10 mM unlabeled methionine. Ribosomes were affinity-purified from cell pellets drop-frozen in liquid nitrogen by resuspending in buffer A and lysing cells with glass beads using a Fast-Prep. Elution

with TEV protease was done in the presence of 2 mM puromycin, following which eluates were bound to recombinant Gst-Dsk2 UBA resin. Washed beads were eluted in 2× SDS-PAGE buffer, and aliquots were resolved by SDS-PAGE. Gels were dried and exposed to film (Kodak, Rochester, NY).

## Sedimentation of ribosomes through sucrose cushions

Cell pellets were resuspended in lysis buffer A described above except that the buffering reagent was 20 mM Hepes, pH 7.4, and glycerol was omitted. Lysate was incubated on ice for ten minutes and then spun at 20,000×$g$ for 10 min. Ribosomes were sedimented at 80,000 rpm for 65 min (RP100AT4 rotor in Sorvall RC M120EX ultracentrifuge) through 1.0 M sucrose cushions. Ribosome pellets were resuspended in SDS sample buffer and evaluated by Coomassie blue staining, or immunoboltting with anti-Ub antibodies.

## Acknowledgements

We thank Drs. Tsui-Fen Chou, Andre Gerber, Dale Haines, Ambro van Hoof, Claudio Joazeiro, Ernst Jarosch, Roy Parker, Matthew Sachs, Thomas Sommer, Peter Walter and Jonathan Warner for reagents. David Akopian's help in using the sucrose gradient fractionator in the Shu-Ou-Shan laboratory is gratefully acknowledged. We thank current and ex-members of the Deshaies Lab for fruitful discussions, particularly Drs. Gary Kleiger and Willem den Besten. Finally, we thank Jonathan Weissman and Ramanujan Hegde for sharing data prior to publication.

## Additional information

### Competing interests

RJD: Reviewing Editor, *eLife*; Founder, shareholder, and consultant of Cleave Biosciences, which is developing drugs that target enzymes involved in protein homeostasis. The other authors have declared that no competing interests exist.

### Funding

| Funder | Grant reference number | Author |
|---|---|---|
| Howard Hughes Medical Institute | | Raymond J Deshaies |

The funder had no role in study design, data collection and interpretation, or the decision to submit the work for publication.

### Author contributions

RV, Conception and design, Acquisition of data, Analysis and interpretation of data, Drafting or revising the article; RSO, Acquisition of data; NJK, Acquisition of data; RJD, Conception and design, Analysis and interpretation of data, Drafting or revising the article

## Additional files

### Supplementary files

• Supplementary file 1. Yeast strains and plasmids used in this study.

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
