## [Decision Letter]

Thank you for choosing to send your work entitled “Cdc48/p97 promotes degradation of aberrant nascent polypeptides bound to the ribosome” for consideration at *eLife*. Your submission has been evaluated by three reviewers, two of whom are members of our Board of Reviewing Editors, and the decision has been discussed further with one of *eLife's* Senior Editors. The following reviewer wants to reveal his identity: Roy Parker.

General assessment and substantive concerns: This manuscript addresses the role of Cdc48 in targeting nascent aberrant polypeptides for degradation. The main conclusion is that the Cdc48 complex plays a role in disassembling tRNA-peptide-ubiquitin conjugates from the ribosome. Cdc48 has been previously characterized as a “seggregase” to remodel certain ubiquitinated substrate complexes prior to their proteasomal degradation, but these findings are relevant because they describe a new role for Cdc48 in ribosome-associated protein quality control pathways. All of the reviewers have credited the quality of the work but they also indicated that mechanistic understanding of this new pathway is missing. For example, no evidence is presented that the Cdc48 role in this process is direct and it remains possible that Cdc48 affects other aspects of the cell that indirectly lead to this phenotype. Further detailed analyses of the pathway and a better mechanistic understanding of the process are needed for publication in *eLife*.

---

## [Author Response]

The main changes from the original submission are as follows:

1. The main criticism of the original submission was that there was not sufficient mechanistic insight into how Cdc48–Ufd1–Npl4 promotes degradation of tUNCs and non-stop decay (NSD) pathway substrates that are translated from messages that lack a stop codon. In particular, we provided no evidence to indicate that the role of Cdc48–Ufd1–Npl4 in this process is direct. In a series of email and telephone discussions with the editors, we were advised that providing evidence to support a direct role for Cdc48–Ufd1–Npl4 in RAD might suffice to address the concern that the reviewers raised about mechanism. We have addressed this criticism by showing in the revised manuscript that Cdc48 and Ufd1 were associated specifically with stringently-washed, affinity-purified ribosomes (Figure 1D). We also show that Cdc48 and Ufd1 bound the NSD substrate GFH^NS^, but not the control GFH^Stop^ (Figure 3E), despite the latter being present at much higher levels than the NS reporter.

2. In addition to the new figure panels described above (Figures 1D and 3E), we have added the following additional data panels to address the other criticisms made by the reviewers: (i) input controls for Figure 1B, (ii) evidence that accumulation of tUNCs in *cdc48-3* cells is reversed by expression of wild type Cdc48 but not the ATPase-deficient Q2 mutant (Figure 1C), (iii) an expanded Figure 2B to include a control in which RNAse was added to cell lysate prior to isolation of ribosomes, (iv) a new version of Figure 3A showing the effect of *ubr1Δ* and *ltn1Δ* on accumulation of tUNCs in *cdc48-3* mutants, (v) a new version of Figure 3B showing the effect of Cdc48 pathway mutations on accumulation of the NSD substrate GFH^NS^, including an anti-tubulin loading control, (vi) degradation assays for GFH^NS^ in wild type, *ltn1Δ*, and *ufd1-2* cells and GFH^Stop^ in wild type cells (Figure 3C), (vii) high-resolution sucrose gradient fractionation of lysates of *ltn1Δ* and *cdc48-3* cells expressing PrA^NS^ (Figure 4B; these data replace the sucrose gradient fractionations of GFH^NS^ that were previously shown in Figure 3D), (viii) levels of Gnd1^PTC^ in total cell extracts of various mutants (Figure 6B), and (ix) post-ribosomal supernatants from the sucrose cushion fractionation of cells expressing Gnd1^PTC^ (Figure 6C bottom panel).

3. Because we do not have direct mechanistic data on exactly what Cdc48 is doing to promote degradation of proteins encoded by mRNAs that lack a stop codon, we have attenuated our Discussion regarding the potential role(s) of Cdc48 in this process.

One week prior to the submission of this revised manuscript, a paper was published by Jonathan Weissman's group (Brandman et. al. 2012 Cell 151, 1042-54; http://www.ncbi.nlm.nih.gov/pubmed/23178123; doi:10.1016/j.cell.2012.10.044) that partially overlaps with our own work in a complementary manner. Weissman's group was studying the heat shock response in yeast, which is mediated by the transcription factor HSF1. This led them to discover the Ribosome Quality Control (RQC) protein complex, which contains Rqc1, Tea2, Ltn1, Cdc48, Ufd1, and Npl4, as well as subunits of the 60S ribosomal subunit. Brandman et al show that Rqc1 and Ltn1 are required for association of Cdc48 with RQC, and hence they propose that Rqc1 and Ltn1 help recruit Cdc48 to the ribosome. Mutant *rqc1Δ* cells accumulate a model substrate (GFP–Arg12–RFP) with an internal polybasic tract that is designed to mimic an NSD pathway substrate, and Rqc1 and Cdc48 are shown to co-immunoprecipitate with this substrate. Although Weissman and colleagues propose that Cdc48 is involved in degradation of RQC substrates, no evidence is provided that Cdc48, Ufd1, or Npl4 are required for RQC function. This leaves open several possible interpretations. As but one example, their data are also consistent with the possibility that Cdc48–Ufd1–Npl4 serves as a negative regulator of RQC that blocks RQC from targeting inappropriate substrates, thereby determining the specificity of RQC action. In that sense, the two papers provide a nice complement to each other – Brandman et al describe a mechanism for how Cdc48 is recruited to NSD pathway substrates, whereas our work establishes that Cdc48, Ufd1, and Npl4 are required to release these substrates from the ribosome so that they can be degraded by the proteasome. We have modified our Discussion to incorporate the findings of Brandman et al and how they relate to our own.